# Comprehensive Echocardiography of Left Atrium and Left Ventricle Using Modern Techniques Helps in Better Revealing Atrial Fibrillation in Patients with Hypertrophic Cardiomyopathy

**DOI:** 10.3390/diagnostics11071288

**Published:** 2021-07-17

**Authors:** Elżbieta Wabich, Agnieszka Zienciuk-Krajka, Radosław Nowak, Alicja Raczak, Ludmiła Daniłowicz-Szymanowicz

**Affiliations:** 1Department of Cardiology and Electrotherapy, Medical University of Gdansk, 80-211 Gdansk, Poland; elzbieta.wabich@gumed.edu.pl (E.W.); agnieszka.zienciuk-krajka@gumed.edu.pl (A.Z.-K.); nowyrad@gumed.edu.pl (R.N.); 2Department of Psychiatry, Medical University of Gdansk, 80-211 Gdansk, Poland; alicja.raczak@gumed.edu.pl

**Keywords:** hypertrophic cardiomyopathy, speckle-tracking echocardiography, two-dimensional strain, three-dimensional strain, atrial fibrillation

## Abstract

Atrial fibrillation (AF) is an important arrhythmia in hypertrophic cardiomyopathy (HCM). We aimed to explore whether a complex evaluation of the left ventricle (LV) using modern echocardiography techniques, additionally to the left atrium (LA) boosts the probability of AF diagnosis. Standard echocardiography, 2D and 3D speckle tracking, were performed for LA and LV evaluation in HCM patients and healthy volunteers. Of 128 initially qualified HCM patients, 60 fulfilled included criteria, from which 43 had a history of AF, and 17 were without AF. LA volume index and peak strain, LV ejection fraction, and strains were significant predictors of AF. In addition, 2D global longitudinal strain (GLS) for LV at cut off −16% turned out to be the most accurate predictor of AF (OR 48.00 [95% CI 2.68–859.36], *p* = 0.001), whereas the combination of LA peak strain ≤ 22% and LV GLS ≥ −16% had the highest discriminatory power (OR 76.36 [95% CI 4.13–1411.36], *p* = 0.001). AF in HCM patients seems to be LA as well as LV disease. Revealing lower strain for LV, in addition to lower LA strain, may have an important impact on accurate characteristics of HCM patients with AF history.

## 1. Introduction

Hypertrophic cardiomyopathy (HCM) is the most common myocardial genetic disease, with a 0.2% incidence in the general population [1,2]. Atrial fibrillation (AF) can frequently complicate this pathology and associate with a worse prognosis [3,4,5]. The absence of the left atrial (LA) active phase is a hemodynamic consequence of this arrhythmia. In the situation of rapid heart rate and typical for HCM diastolic dysfunction, AF could significantly impair left ventricle (LV) filling and increase the LV output tract (LVOT) gradient, leading to an intensification of symptoms associated with this arrhythmia [3]. Additionally, AF in HCM patients requires antithrombotic treatment regardless of the CHA_2_DS_2_-Vasc SCORE [4,5,6,7]. Therefore, the knowledge concerning the history and risk of AF in HCM patients is of great clinical value [4,8]. Thus far, only LA diameter (greater than 45 mm) has been mentioned in cardiological guidelines as the only risk factor recommended in revealing AF in HCM patients [7]. The sensitivity and specificity of this parameter are too low; therefore, other indices of LA are intensively assessed in this issue [5,8,9,10,11]. For instance, it has been suggested that the risk for AF might be better reflected by 2-dimensional (2D) assessed LA volume rather than LA diameter [9,10]. Recent studies postulated speckle tracking echocardiography (STE) as a novel method for a more accurate assessment of LA function than LA size or volume [5,11,12,13,14,15,16,17].

Recently, the evaluation of LV function, not only LA, in AF prediction enjoys growing interest. The connection between the LV and LA functions is well known. The LA reservoir function (consisted of active relaxation and passive dilatation) [16] is associated with the LV systolic contraction, while the conduit LA function and booster pump support and depends on the LV diastolic role [15,16]; therefore, the impairment in LV short-axis shortening and diastolic relaxation could affect the LA function [15,16]. That seems to be particularly important in HCM patients, where the pathophysiology of AF due to complex myocardial mechanics of LV and LA is undoubtedly different and more complex than in persons without this cardiomyopathy [18]. Therefore, the role of LV assessment, in addition to LA, in AF prediction in HCM patients seems to be of great importance. In this issue, 2D STE is a well-proven technique, confirmed for the prediction of malignant ventricular arrhythmias [19,20,21,22,23,24,25], exacerbation of heart failure [21,23,24], and revealing the areas of myocardial fibrosis [21,26]; however, almost none of them were dedicated to the role of LV strains in AF probability.

Moreover, strains calculated in three-dimensional (3D) STE could have a complementary role in HCM patients’ assessment due to the potential ability to overcome some of the intrinsic limitations of 2D STE in assessing complex LV myocardial mechanics offering additional deformation parameters. However, it has been studied so far only in a few publications [27,28,29,30]; none of them were dedicated to HCM patients with AF.

Our study aimed to explore the value of LV assessment, using modern 2D and 3D echocardiography techniques in revealing the history of AF in HCM patients. We tried to answer whether the through LV assessment additionally to LA boosts the probability of AF diagnosis.

## 2. Materials and Methods

### 2.1. Study Population

A prospective analysis of retrospectively enrolled consecutive patients of the Outpatient University Clinic in Gdansk with HCM diagnosis was performed with data collected from October 2013 to October 2018. The HCM diagnosis was based on the current criteria: determined by echocardiography, at least one or more LV segmental hypertrophy ≥ 15 mm, which could not result from abnormal loading condition (such as hypertension or aortic stenosis) [7]. The AF was diagnosed based on the information in medical documentation (including Holter-monitoring performed in every patient before the enrollment as the routine procedure in HCM patients in our outpatient clinic). The exclusion criteria were permanent and persistent atrial fibrillation, severe valvular defects, history of myectomy or alcohol septal ablation, history of AF or supraventricular arrhythmias ablation, and a severe general condition from comorbidities insufficient parameters of the echocardiographic images unable to perform the 2D and 3D STE analysis, and age below 18 years. Sex and age-matched healthy volunteers constituted the control group (as a reference group for the STE measurements). The study’s protocol was approved by the local ethics committee of the Medical University of Gdansk (NKBBN/390/2018), and written informed consent was obtained from all participants.

### 2.2. Echocardiography

Echocardiography examination was performed during sinus rhythm, using the GE VIVID E9 machine (GE ultrasound System, Horten, Norway), equipped with phased-array transducer (M5S) and 3D (4V) probe. In addition, 2D echo sets were obtained following applicable recommendations using parasternal and apical views [31]: the images were recorded in three consecutive heart cycles during quiet respiration.; grayscale recordings were optimized at a frame rate of 50–80 frames/s. Immediately after 2D, 3D echocardiography was performed under the same hemodynamic conditions, from the apical position in four-chamber view, according to applicable recommendations [31]; the recordings were performed over six cardiac cycles (multi-beat acquisition) in order to obtain the lowest speed of 20 images per second, when the subject had to hold his breath, most commonly in deep expiration. All echocardiograms were stored digitally, and further offline analysis was performed using a commercial EchoPAC workstation (v202, GE Healthcare, Horten, Norway). Two experienced echocardiographers performed all echocardiographic analyses.

#### 2.2.1. Standard Echocardiography Parameters

LA diameter size (LADs) was measured in the parasternal view. The LA and LV volumes and LVEF were measured by Simpson’s Biplane Method in the apical 2D views. LV mass was calculated as it was described in the literature [31]. The LA volume and LV mass were indexed by body surface area (BSA). The mitral inflow velocity was obtained by placing a pulsed-wave Doppler sample volume above the mitral leaflet tips during diastole from the apical 4-chamber view. The peak early (E) and late (A) transmitral flow velocities and deceleration time of the E velocity (DT) were measured. The ratio of early-to-late peak velocities (E/A) was calculated. Tissue doppler imaging was performed to measure the mitral annulus excursion velocity. A pulsed-wave sample volume was placed at the lateral and septal corners of the mitral valve annulus. The early diastolic (Em) myocardial peak velocity was recorded and averaged from both positions E/Em ratio was calculated [31]. Right ventricle internal diameter (RVID) was obtained and measured at end-diastole from the RV-focused 4-chamber view in the basal of RV inflow [31].

#### 2.2.2. 2D STE Parameters

2D STE parameters were obtained following the applicable recommendations [32]. For the longitudinal STE analysis of LV, three endocardial markers were placed in an end-diastolic frame in the apical four-, two-, and three-chamber views. The software automatically tracked the contour of the endocardium to cover the entire LV wall’s myocardial thickness. Adequate tracking could be verified in real time and corrected by adjusting the region of interest or manually to ensure optimal tracking. Segments with low tracking quality were not taken into account in appropriate STE analysis; if the patient had three or more segments with poor tracking quality, the software automatically did not allow for further analysis. Then, 2D peak systolic longitudinal strain (a global longitudinal strain of the LV-LV GLS) was analyzed from the apical views and calculated for the 16 from 17 segments (six basal, six mid, and four apical) concerning the strain magnitude at the aortic valve closure [33]. Due to the absence of software dedicated to LA strain analysis, the peak-atrial longitudinal strain was measured as the average value from two- and four-chamber views by using LV strain software, according to the methodology described in previous studies [34,35]; the peak atrial longitudinal strain was obtained during the ventricular systole with a QRS complex as a reference point [36]. LV twist was obtained as the highest difference in degrees between the apical and basal rotation. LV torsion was defined as LV twist indexed by LV diastolic longitudinal length (the distance between the mitral annulus and the apex in end-diastole averaged from four-, two-, and three-chamber apical views) [26].

#### 2.2.3. 3D STE Parameters

3D measurements were obtained following applicable recommendations [37,38]. The four-chamber plane was manually pivoted to align the three apical views to track the intersection line in the middle of the LV cavity in every (two-, three- and four-) apical view by crossing the LV apex and central place of the mitral valve. The software’s endocardial LV cavity was automatically detected, corrected manually if needed to provide the measured LV volumes. In the next step, the epicardial border was automatically detected by the software and corrected manually if needed; then, 3D STE analysis was performed on that basis, 3D area strain of the LV (LV area strain—calculated automatically adding longitudinal and circumferential strains), and 3D radial strain of the LV was acquired [37].

### 2.3. Statistics

Continous data are presented as median (25th percentile–75th percentile), whereas categorical data are expressed as proportions. The Shapiro–Wilk test was performed to determine whether data were normally distributed. The majority of the analyzed parameters did not have a normal data distribution, even after logarithmic data transformation. Therefore, we selected appropriate statistical analysis methods based on nonparametric tests. Comparisons between groups were performed with the Mann–Whitney U-test for continuous variables and Pearson’s chi-square test for categorical variables, as appropriate. Intra- and inter-observer reproducibility was assessed on 20 randomly selected patients: intra-class correlation coefficient (ICC), coefficient of variation (CV), the lower and upper limits of agreement, and mean bias (Bland–Altman test) were calculated. The clinical significance of ICC was interpreted as follows: excellent, ICC ≥ 0.80; good, 0.60 ≤ ICC < 0.80; moderate, 0.40 ≤ ICC < 0.60; and poor, ICC < 0.40 [39]. The accuracy of echocardiography parameters as potential predictors of AF was determined using the area under the receiver-operating characteristic (ROC) curve (AUC). Univariate logistic regression analysis was performed to detect which indices (with prespecified cutoff values) show the most substantial relation to the presence of AF. Values of *p* < 0.05 were considered significant. The statistical analysis was conducted with and R 3.1.2. environment (R Core Team, Vienna, Austria).

## 3. Results

### Study Population

Amongst 128 consecutive HCM patients, finally, we enrolled 60 (Figure 1), from which 43 had a history of AF (AF+ group); and 17 patients were without AF diagnosis (AF− group). The control group consisted of 23 healthy volunteers. The AF+ and AF− groups significantly differed in the 5-year risk of sudden cardiac death and implantable cardioverter defibrillator rate but did not vary in concomitant diseases and essential pharmacological treatment (Table 1).

Among standard echocardiography, LADs, LAA index, LAV index, and diastolic function parameters (Em and E/Em) differed significantly between HCM patients and healthy persons; regarding the comparison between AF+ and AF− patients, the only parameter significantly differed between the two groups was LVEF, which was lower in AF+ patients (Table 2). The accurate measurement of the 2D LV GLS was able to perform in every included patient; the reliable analysis of LA strain was possible in 43 patients, whereas 3D LV strains were accurately measured in 45 patients. Most of the advanced echocardiographic parameters regarding LA and LV were worse in HCM patients than in healthy volunteers. Comparing AF+ and AF− groups, strain parameters for LV were significantly lower in AF+ patients (Table 2). Figure 2, Figure 3, Figure 4 and Figure 5 represent the examples of LA and LV strain analysis in HCM patients with FA diagnosis, without a history of FA, and a healthy person.

Prespecified cut-off values calculated in the ROC analysis are presented in Table 3. Strains for LV (based on 2D and 3D STE) turned out to be the most accurate predictors of AF occurrence, whereas other parameters had lower discriminatory power. Figure 6 presents the most essential ROC curves for 2D and 3D STE parameters.

LAVi, LA peak strain, LV strains, and LVEF with prespecified cut-off values calculated from the ROC curves were significant predictors of AF in the univariate logistic regression analysis that was not confirmed for LADs (Table 4).

LV GLS turned out to be the strongest predictor of AF: OR 48.00 (2.68–859.36), *p* = 0.001. A combination of LV GLS worse than −16% and LA peak strain lower than 22% in logistic regression analysis increased the probability of AF revealing OR 76.36 (4.13–1411.36), *p* = 0.001.

The inter- and intra-observer variability analysis shows similar and satisfactory results for the measured STE parameters: for 2D LA and LV strains and 3D LV strains (Table 5). The ICC was excellent (in every measured value higher than 0.8). For intra-observer and inter-observer analyses, the CVs were highest for 2D LA peak strain (19.2% and 15.6%, respectively), whereas the smallest CVs in intra- and inter-observer variabilities were obtained in 3D area strain analysis (11.7% and 12.8%, respectively).

## 4. Discussion

The main finding of the present study is revealing the complex mechanism of AF in HCM patients based not only on the LA dysfunction but the LV dysfunction as well. We confirmed that the measurement of the LV strains additionally to LA boosts the probability of AF diagnosis. To the best of our knowledge, this is the first study where a thorough analysis of the LV, including 2D and 3D speckle-tracking techniques, additionally to LA assessment, was performed in AF diagnosis revealing in HCM patients.

LADs, mentioned in the European HCM guidelines [7] as the only parameter demanding the active search for AF in HCM patients, in the presented study turned out to be a parameter of low specificity (60%), with the AUC value of only 63.7%, and not statistically significant in the univariate logistic regression analysis (*p* = 0.078). LAVi, as the following standard echocardiography parameter, turned out to be better than the LADs predictor of AF history, being in line with data from the literature [5,8,9,10,11]. It is worth noting that LA minimal volume (compared to standard used LA maximal volume) in some previous studies is postulated as a more sensitive parameter to reveal subtle LA fibrotic changes [40]. However, in our research, the HCM patients had somewhat bigger than smaller LA size and volume parameters, as it is presented in Table 2. Better predictive accuracy for discrimination of the AF patients in our study was revealed for LA strain. It is worth noting that the pre-specified cut-off value for LA strain in our study was similar to data from the literature [5,16]: in Debonnaire P et al. study [5], that was 23.4%, whereas, in Fujimoto K. et al. [16], the value constituted 20.3%. This parameter turned out to be a significant predictor in the univariate logistic regression analysis was higher than for LAVi predictive power: OR 17.11 (0.95–308.16), *p* = 0.005.

Our study revealed the differences in the LV strains between the HCM patients with and without AF (Table 2), postulating that AF in HCM patients could be the LA as well as the LV disease. In our research, LV longitudinal strain measured in 2D was the most powerful predictor of AF with the highest level of the odds ratio (Table 4). Our results would postulate that this parameter with the cut-off value worse than −16% could be supplementary in revealing HCM patients with AF history; moreover, the combined analysis of 2D strains for LA and LV (Table 4) could increase the probability of AF revealing and seems to be more valuable than using each of these parameters separately. Several studies highlighted the complex and unique mechanisms of rotation and twist and the role of those parameters in LV function in HCM patients [26,41]. However, in our study, these parameters did not differ between AF+ and AF− patient, which could be the consequence of small sample size; therefore, it is impossible to say that these parameters are not related to the risk of AF in HCM patients and should be verified in the further, bigger studies.

To the best of our knowledge, our research is one of the first studies concerning the relationship between the LV strain and AF occurrence in patients with HCM. Russo et al. in the Northern Manhattan Study of 675 patients showed that a value measured from 2D GLS for LV worse than –14.7% was a strong predictor of AF appearance [42], but this study was not concentrated on patients with HCM. Zegkos et al., in his recently published research concerning the relationship between 2D LV and LA strains and AF occurrence in the HCM population, defined the optimal cutoff values for the LV GLS (≥−14%) and LA strain (<20%) [43], which are very similar to that presented in our study.

Our study’s innovation is that 3D STE for the LV evaluation may help diagnose AF in HCM patients. Indeed, 3D echocardiography plays a particular role in assessing HCM patients due to the complex mechanics of LV contraction [27,29,44] (Table 3). Tanaka et al. [45] suggested that only 3D STE analysis could provide reliable and accurate information about proper LV mechanics. Pagourelias et al. [30] noted that 3D speckle tracking could have a potential role in reflecting LV fibrosis in HCM patients. Our study is the first to describe the 3D STE parameters in diagnosing AF history in HCM patients. Although the discriminatory power of prespecified values for the 3D area and radial strain was lower than for the 2D longitudinal strain (Table 4), the abovementioned findings could highlight the potential usefulness of 3D strain in this issue.

### 4.1. Clinical Implications

The knowledge of the occurrence and early diagnosis of AF in HCM patients is of great clinical value. Widely available and easy to perform, STE may be helpful in the identification of HCM patients with a high risk of AF appearance. Strain for the LA enjoys a growing clinical interest in this issue. GLS for LV measured by 2D STE is a well-studied, validated, and accessible parameter, useful in many clinical conditions, including HCM patients. Diagnosis of AF in this group could be another essential clinical application for this parameter, particularly in combination with the LA strain measurement. Furthermore, 2D LV GLS worse than −16%, in addition to 2D LA strain below 22%, allows for AF diagnosis with high probability. Moreover, 3D STE analysis opens up further clinical possibilities in AF risk estimation of HCM patients.

### 4.2. Study Limitations

Due to the necessity of selecting a clinically homogeneous group of patients, the most critical limitation is that this study is a single-center analysis based only on a small group of patients with taking into account only patients with good echocardiographic image quality, especially considering the evaluation of strain. The small number of patients translated to the lack of multivariate analysis and the lack of other kinds of statistics (for instance, assessing the added-value of the 2D and 3D strains over the well-known classical parameters by using the net reclassification index). Contrary to complex 3D STE parameters, we did not analyze 2D circumferential and radial strains as an analysis that requires a complex approach. The next limitation is that LA strain was analyzed using the software for LV strain analysis due to the lack of appropriate software dedicated to LA. Furthermore, GLS was assessed using only one of the available vendor platforms; therefore, the results may differ slightly from those received with alternative software algorithms.

## 5. Conclusions

AF in HCM patients seems to have a complex mechanism based on LA and LV dysfunction. Modern echocardiography techniques could help for better discrimination of HCM patients with a history of AF, which may have a potential clinical impact due to the necessity of anticoagulation treatment in such patients regardless of the other known risk factors for thromboembolic complications. Further research based on a more extensive group of patients is strongly recommended.

## Figures and Tables

**Figure 1 diagnostics-11-01288-f001:**
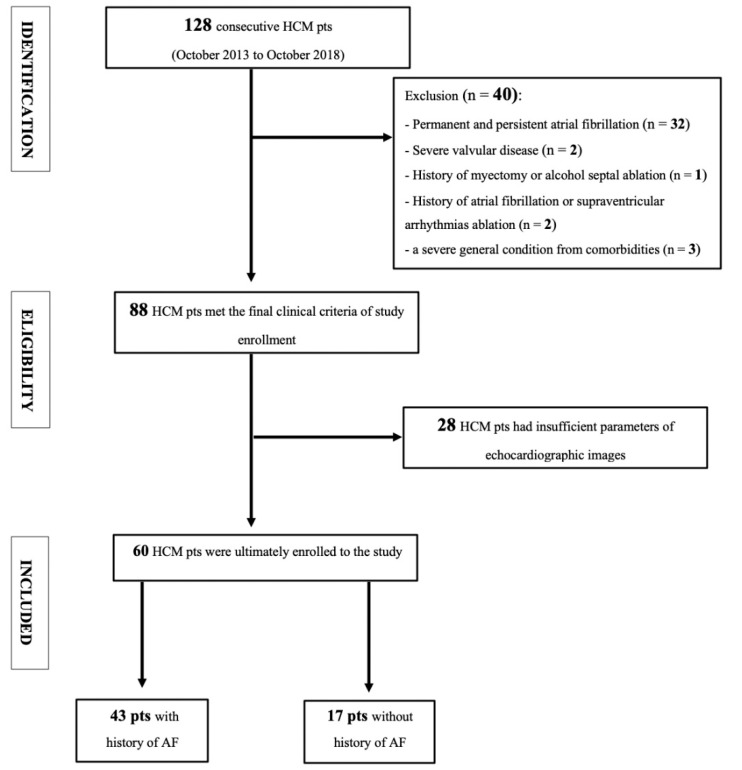
Flow chart of screened, included, and excluded HCM patients.

**Figure 2 diagnostics-11-01288-f002:**
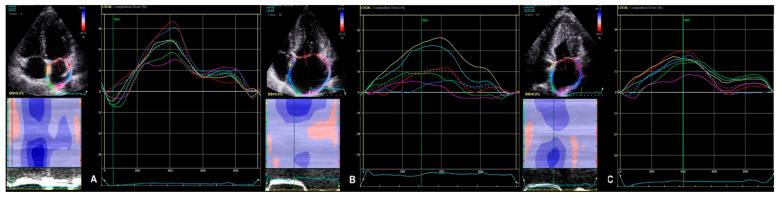
The example of 2D LA strain analysis derived from a four-chamber apical view of a healthy person (**A**); HCM patient with FA diagnosis (**B**); HCM patient without FA diagnosis (**C**).

**Figure 3 diagnostics-11-01288-f003:**
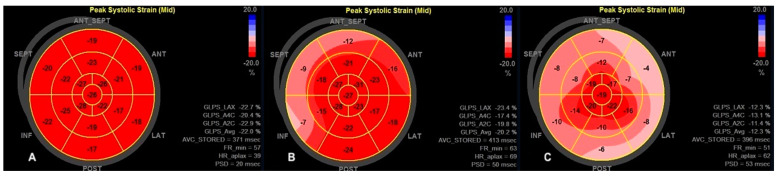
The example of 2D LV global longitudinal strain measurements—the bull eye representation of a healthy person (**A**); HCM patient without FA diagnosis (**B**); HCM patient with FA diagnosis (**C**).

**Figure 4 diagnostics-11-01288-f004:**
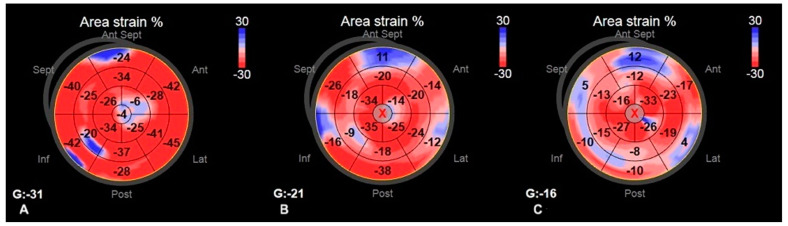
The example of 3D LV area strain measurements—the bull eye representation of a healthy person (**A**); HCM patient without FA diagnosis (**B**); and HCM patient with FA diagnosis (**C**).

**Figure 5 diagnostics-11-01288-f005:**
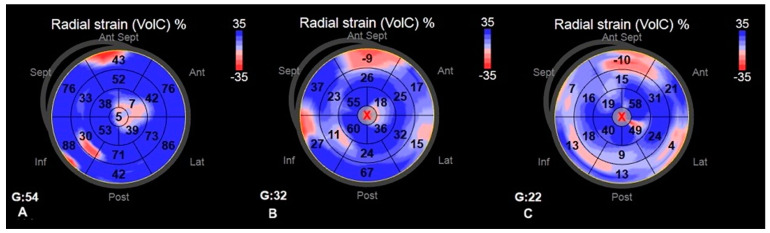
The example of 3D LV radial strain measurements—the bull eye representation of a healthy person (**A**); HCM patient without FA diagnosis (**B**); HCM patient with FA diagnosis (**C**).

**Figure 6 diagnostics-11-01288-f006:**
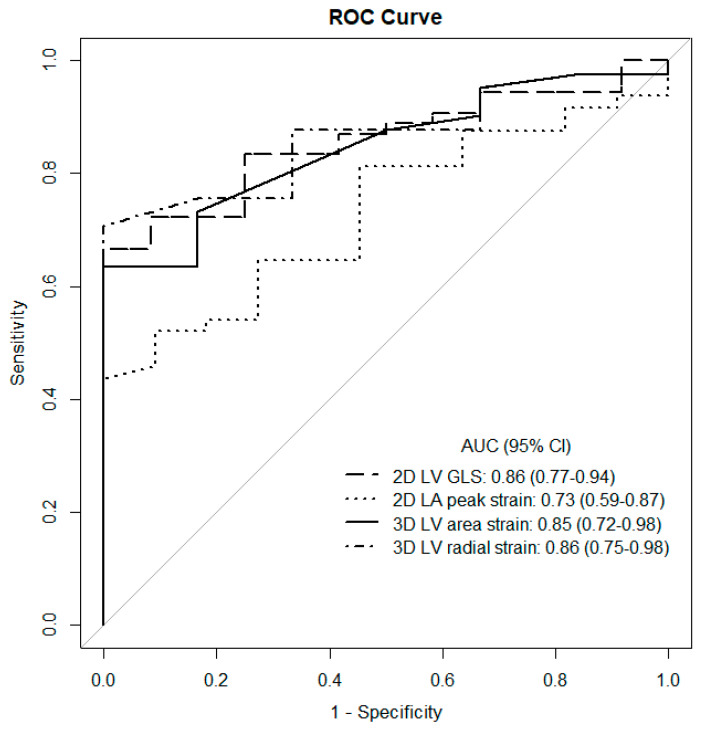
ROC curves of 2D and 3D STE parameters.

**Table 1 diagnostics-11-01288-t001:** Clinical characteristics of HCM AF+ and AF− patients.

	HCM AF+ (*n* = 43)	HCM AF− (*n* = 17)	*p*
Age	57 (45.5–63)	56 (41–63)	0.381
SCD in family history *n* (%)	7 (16)	6 (35)	0.163
5-year risk of SCD *n* (%)	3.6 (2.6–4.9)	5.5 (3.5–9.4)	<0.030
History of non- sustained ventricular tachycardia, *n* (%)	9 (21)	7 (41)	0.195
History of syncope/ presyncope *n* (%)	17 (40)	8 (47)	0.772
Implantable cardioverter—defibrillator, *n* (%)	7 (16)	8 (47)	<0.021
**Concomitant diseases**			
Hypertension *n* (%)	25 (58%)	6 (35%)	0.154
Coronary artery disease *n* (%)	3 (7%)	0 (0%)	0.551
Myocardial infarction *n* (%)	1 (2%)	0 (0%)	1.000
Diabetes mellitus type 2 *n* (%)	8 (19)	2 (12)	0.709
Hyperlipidemia *n* (%)	26 (62)	11 (65)	1.000
Smoking *n* (%)	16 (37)	6 (35)	1.000
**Medications**			
Beta-blockers, *n* (%)	32 (48)	16 (94)	0.151
ACE inhibitors, *n* (%)	19 (44)	7 (41)	1.000
Spironolactone *n* (%)	1 (2)	1 (6)	0.490
Calcium—blocker *n* (%)	6 (14)	2 (12)	1.000
Cordarone/Sotalol *n* (%)	3 (7)	3 (12)	0.338
Diuretics *n* (%)	10 (23)	2 (12)	0.479
Statins *n* (%)	19 (44)	8 (47)	1.000

SCD: sudden cardiac death.

**Table 2 diagnostics-11-01288-t002:** Echocardiographic parameters.

	HCM All *n* = 60	Healthy Volunteers *n* = 23	*p* HCM vs. Healthy	AF+ *n* = 43	AF− *n* = 17	*p* AF+ vs. AF−	*p* AF+ vs. Healthy	*p* AF− vs. Healthy
**Standard echocardiography parameters**								
LADs	45 (42–48)	38 (35–40)	<0.000	45 (42–48)	45 (42–49)	0.474	<0.000	<0.002
LAVi	53 (42–62)	27 (23–32)	<0.000	58 (45–63)	51 (41–61)	0.162	<0.000	<0.000
LVMi	178 (149–215)	69 (60–76)	<0.000	198 (152–218)	155 (140–172)	0.061	<0.000	<0.000
E/A	0.97 (0.77–0.04)	1.28 (0.94–1.57)	0.073	0.94 (0.76–1.04)	1.23 (0.79–1.04)	0.357	0.066	0.242
DecT	185 (150–263)	182 (149–214)	0.216	182 (149–268)	208 (167–240)	0.352	0.318	0.122
Em	0.06 (0.05–0.08)	0.12 (0.09–0.14)	<0.000	0.06 (0.05–0.08)	0.06 (0.05–0.07)	0.347	<0.000	<0.000
E/Em	10.7 (9.0–14.5)	6.8 (5.4–8.0)	<0.000	10 (8.5–14.9)	11.5 (9.8–12.2)	0.374	<0.000	<0.000
LVEF	64 (55–69)	63 (61–63)	0.098	53 (47–64)	67 (61–70)	<0.010	0.052	<0.008
EDV	90 (71–121)	108 (89–126)	0.055	90 (71–116)	101 (70–125)	0.453	<0.049	0.202
ESV	35 (23–45)	40 (33–47)	<0.037	31 (21–44)	41 (27–58)	0.069	<0.027	0.421
RVID	25 (23–27)	26 (22–29)	0.142	25 (22–27)	25 (24–26)	0.353	0.128	0.306
Speckle tracking echocardiography parameters								
2D LA peak strain	15.9 (12.3–20.0)	28.5 (22.4–31.3)	<0.000	13.8 (10.6–18.4)	16.5 (12.9–21.5)	0.164	<0.000	<0.000
2D LV GLS	−15.2 (−17.5–−12.1)	−19.6 (−20.9–−17.9)	<0.000	−12.4 (−14.2–−10.0)	−16.3 (−19.1–−13.9)	<0.003	<0.000	<0.001
3D LV area strain	−23.0 (−26.5–−21.0)	−26.5 (−28.8–−20.0)	0.109	−19.5 (−20.8– −17.5)	−25.0 (−27.0–−22.0)	<0.009	<0.015	0.282
3D LV radial strain	35.0 (29.5–42.5)	39.5 (−29.0–−46.8)	0.146	27.7 (24.8– −30.3)	37.0 (32.0–−44.0)	<0.006	<0.011	0.364
2D Twist	21.2 (16.3–25.5)	24 (19.8–26.3)	0.063	16.75 (12.54–22.06)	22.01 (17.25–25.45)	0.165	<0.036	0.126
2D Torsion	2.9 (2–3.4)	3 (2.7–3.9)	0.059	2.4 (1.8–3)	3 (2.3–4)	0.198	<0.036	0.121

HCM: hypertrophic cardiomyopathy; AF: atrial fibrillation; LADs (mm): left atrium diameter size; LAVi (ml/BSA): indexed left atrium volume; LVMi (g/BSA): indexed left ventricle mass; E/A: the ratio of early to-late peak mitral velocities; DecT (msec): deceleration time; Em (cm/s): early diastolic mitral myocardial peak velocity averaged from the lateral and septal positions; E/Em: the ratio between E and Em; EDV (ml): the end-diastolic volume of the left ventricle; ESV (ml): the end-systolic volume of the left ventricle; RVID (mm): right ventricle internal diameter; LVEF (%): left ventricle ejection fraction; LV GLS (%): 2D global longitudinal strain of the left ventricle; LA peak strain (%): 2D peak longitudinal strain of the left atrium; LV area strain (%): 3D left ventricular area strain; LV radial strain (%): 3D left ventricular radial strain; 2D Twist (°): left ventricular twist; 2D Torsion (°/cm): left ventricular torsion.

**Table 3 diagnostics-11-01288-t003:** Cut-off values and prognostic accuracy of the analyzed 2D and 3D STE parameters (calculated in the ROC analysis) as predictors of AF presence.

Parameter	AUC	Characteristics (95% CI)		Predictive Value (95% CI)	
		Sensivity (%)	Specificity (%)	Positive (%)	Negative (%)
LADs ≥ 45 mm	63.7	67	60	88	30
LAVi ≥ 57 mL/m^2^	75.9	79	67	91	42
LVEF ≤ 55%	70.4	89	60	89	60
LA peak strain ≤ 22%	73.0	44	100	100	29
LV GLS ≥ −16%	85.6	67	100	100	40
LV area strain ≥ −23.5%	84.6	63	100	100	29
LV radial strain ≤ 33%	86.4	71	100	100	33

AUC: area under curve; CI: confidence interval.

**Table 4 diagnostics-11-01288-t004:** Univariate logistic regression analysis as a predictor of the composite endpoint.

Parameter	Univariate Analysis	
	OR (95% CI)	*p*
LADs ≥ 45 mm	3.00 (0.94–9.55)	0.078
LAVi ≥ 57 mL/m^2^	7.45 (1.89–29.41)	0.004
LVEF ≤ 55%	12.25 (3.22–46.61)	0.001
LA peak strain ≤ 22%	17.11 (0.95–308.16)	0.005
LV GLS ≥ −16%	48.00 (2.68–859.36)	0.001
LV area strain ≥ −23.5%	20.80 (1.08–399.06)	0.005
LV radial strain ≤ 33%	29.00 (1.5–560.98)	0.002
LV GLS ≥ −16% and LA peak strain ≤ 22%	76.36 (4.13–1411.36)	0.001

OR: odds ratio; CI: confidence interval.

**Table 5 diagnostics-11-01288-t005:** Inter- and intra-observer variability of 2D LA, LV, and 3D LV strain values.

Parameter	Intra-Observer				Inter-Observer			
	Bias	95% Limits of Agreement	ICC	Coefficient of Variation (%)	Bias	95% Limits of Agreement	ICC	Coefficient of Variation (%)
2D LA peak strain	−0.8	−7–5.5	0.8 *	19.2	−0.6	−5.9–4.6	0.8 *	15.6
2D LV GLS	0.6	−3.9–5.1	0.8 *	14.6	−0.4	−5.2–4.4	0.8 *	16.2
3D area strain	0.4	−4.9–5.6	0.9 *	11.7	0.5	−5.4–6.3	0.9 *	12.8
3D radial strain	−1.4	−13.8–11.1	0.9 *	17.6	−0.7	−11.5–10.2	0.9 *	15.0

ICC—intra-class correlation coefficient. * *p* < 0.001.

## Data Availability

The data sets used and/or analyzed during the current study are available from the corresponding author on reasonable request.

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
