# Peer review of "Comprehensive Echocardiography of Left Atrium and Left Ventricle Using Modern Techniques Helps in Better Revealing Atrial Fibrillation in Patients with Hypertrophic Cardiomyopathy"

_diagnostics, 2021, doi:10.3390/diagnostics11071288_

Round 1
Reviewer 1 Report
Authors aimed to explore whether an evaluation of the left ventricle (LV), additionally to the left atrium (LA), using modern echocardiography techniques, boosts the discriminatory power of atrial fibrillation (AF) in hypertrophic cardiomyopathy (HCM). Standard echocardiography, 2D and 3D speckle tracking, were performed for LA and LV evaluation in HCM patients and healthy volunteers.
From 128 initially qualified HCM patients 60 fulfilled included criteria, from which 43 had a history of AF and 17 were without AF. LA volume index and peak strain, LV ejection fraction and strains, were significant predictors of AF. 2D global longitudinal strain (GLS) for LV at cut-off -16% turned out to be the most accurate predictor of AF, whereas the combination of LA peak strain ≤ 22% and LV GLS ≥ - 16% had the highest discriminatory power. They concluded that revealing lower strain for LV, additionally to lower LA strain, may have an important impact on accurate characteristics of HCM patients with AF history.
The following criticisms would strengthen this manuscript:
- Page 3, line 140. How did you define intraobserver and interobserver variability?
- Page 4, line 153. Which was the feasibility of strain and volume parameters?
- Page 4, line 160. Do you have data on intraobserver and interobserver variability?
- Page 8, line 207. The novelty of the study should be pointed out compared to previous studies.
- Page 9, line 216. The role of minimal LA volume compared to maximal volume should be briefly discussed (ref. Ultrasound Med Biol 2018;44:1198-1211).
- Page 9, line 225. Some points on pathophysiology should be clarified (relation between LA dysfunction and LV diastolic dysfunction and between LA reservoir function and conduit function).
- Page 9, line 247. A specific paragraph on clinical implications should be discussed.
Author Response
Response to the First Reviewer
The authors wish to thank for professional and most valuable comments
provided by the Reviewer. The Reviewer introduced critical new aspects and allowed for
the authors to approach their data with ample criticism. The authors took care to answer all
suggestions and remarks with due attention and diligence, to the limit of their study data.
- How did you define intraobserver and interobserver variability?
The authors wish to thank for this valuable comment. We added the appropriate information in the Statistics section (page 4, lines: 242-246), and included the results (Results section, page 10, lines: 355-360, Table 5).
- Which was the feasibility of strain and volume parameters?
Regarding the remark concerning the feasibility of strain parameters, the authors wish to thank to Reviewer for this just comment.
In our analysis, 88 HCM patients met the final clinical criteria; unfortunately, 28 of them had to be excluded due to the poor echogenicity enabling to perform speckle tracking analysis. The 3D - STE analysis requires good quality echocardiographic images more restrictive due to the necessity for an appropriate visualization of the hypertrophied myocardium for optimal tracking. The accurate measurement of the 2D LV GLS was able to perform in every included patient; the reliable analysis of LA strain was possible in 43 patients (due to the fact that we must excluded from our analysis LA segments with pulmonary veins and patients with interatrial septal aneurysm), whereas 3D LV strains were accurately measured in 45 patients.
Moreover the feasibility of the volume parameters was possible in all included to analysis patients.
We added this valuable information in the Results section (page 6, lines: 273-275).
- Do you have data on intraobserver and interobserver variability?
The authors wish to thank for this valuable comment. We added the appropriate analyses concerning the intra- and interobserver variability in the Results section (page 10, lines: 355-360); also, the precise results presents Table 5
- The novelty of the study should be pointed out compared to previous studies.
The authors wish to thank Reviewer for this valuable remark. We appreciate the reviewer’s insightful suggestion and add an appropriate description to emphasize the novelty of our study, especially underlining that this is the first study where a thorough analysis of the LV, including 2D and 3D speckle tracking techniques, additionally to LA assessment was performed in AF diagnosis revealing in HCM patients. We added precise information in the Introduction (page 2, lines 113-115, 117-118, 122-123) and Discussion section (page 11, lines: 419, 406-408, 427-428, 432-436). We hope that the Reviewer will accept this answer.
- The role of minimal LA volume compared to maximal volume should be briefly discussed (ref. Ultrasound Med Biol 2018;44:1198-1211).
The authors wish to thank Reviewer for raising that important point here. The LA minimal volume (compared to standard used LA maximal volume) in some previous studies is postulated as a more sensitive parameter to reveal subtle LA fibrotic changes, which is very important from the clinical point of view. Contrary, in our study, HCM patients were characterized as somewhat bigger than smaller LA size and volume (what is presented in comparison to the healthy persons in Table 2). According to the Reviewer’s suggestion and the clinical importance of that comment, we added this information and discussed it (Discussion part, page 10, lines 375-379).
- Some points on pathophysiology should be clarified (relation between LA dysfunction and LV diastolic dysfunction and between LA reservoir function and conduit function).
The authors wish to thank for this valuable remark. We add appropriate clarification in Introduction section (page 2, lines 106- 113) and now it reads as follow: “Recently, the evaluation of LV function, not only LA, in AF prediction enjoys growing interest. The connection between the LV and LA function is well-known. The LA reservoir function (consisted of active relaxation and passive dilatation) [16] is associated with the LV systolic contraction, while the conduit LA function and booster pump support and depends on the LV diastolic role [15-16] therefore the impairment in LV short-axis shortening and diastolic relaxation could affect the LA function [15-16]. That seems to be particularly important in HCM patients, where the pathophysiology of AF due to complex myocardial mechanics of LV and LA is undoubtedly different and more complex than in persons without this cardiomyopathy [18].” We hope that this answer will be accepted by the Reviewer.
- A specific paragraph on clinical implications should be discussed.
The authors are very grateful for the Reviewer for this insightful remark. We add the Clinical Implications section (page 11, lines 437-447) according to The Reviewers suggestion and now it reads as follow: “The knowledge of the occurrence and early diagnosis of AF in HCM patients is of great clinical value. Widely available and easy to perform, STE may be helpful in the identification of HCM patients with a high risk of AF appearance. Strain for the LA enjoys a growing clinical interest in this issue. GLS for LV measured by 2D STE is a well-studied, validated, and accessible parameter, useful in many clinical conditions, including HCM patients. Diagnosis of AF in this group could be another essential clinical application for this parameter, particularly in combination with the LA strain measurement. 2D LV GLS worse than -16% additionally to 2D LA strain below 22% allows for AF diagnosis with high probability. Moreover, 3D STE analysis opens up further clinical possibilities in AF risk estimation of HCM patients.” The authors hope this section will be acceptable for the Reviewer.

Reviewer 2 Report
In this retrospective single-center study including 60 patients with hypertrophic cardiomyopathy (HCM) (43 with atrial fibrillation (AF) and 17 without) between October 2013 and October 2018 and 23 healthy volunteers, the authors analyzed 2D and 3D speckle tracking for left atrial (LA) and left ventricle (LV) evaluation. The authors found that LA volume index and peak strain, LV ejection fraction and strains were significant predictors of AF. A 2D-GLS for LV at -16% was the best threshold value to predict AF. The authors concluded that AF in HCM patients seemed to be not only LA but also LV disease. The authors are to be congratulated for their work and this interesting study, using innovative echocardiographic parameters of LA and LV function. Nonetheless, there are many concerns that need to be discussed. Please consider the following comments.
1. The main goal of the study in the abstract (and in the manuscript) is not clear. What does “the discriminatory power of this arrhythmia” mean? Please clarify.
2. The following sentence in the introduction is not clear: “Therefore, the role of LV assessment in this issue, not only the LA, cannot be overestimated”. Please clarify.
3. Please describe what RVID is and how you have obtained this parameter.
4. Please clearly indicate the inclusion criteria of patients with HCM. When were the healthy volunteers included? Why did you include healthy volunteers? I do not quite understand what the added-value of the healthy volunteers is in your analysis. I would suggest removing all data concerning the healthy volunteers. Please justify and clarify this point.
5. How many patients were excluded because of poor echogenicity, preventing SPE analysis? Please also indicate in the manuscript and in Figure 1 the proportion of patients in whom 2D and 3D-SPE analysis was not possible.
6. Regarding SPE measurement, you have used the values automatically displayed by the software. How did you take into account the segments that were not adequately tracked (because of poor echogenicity for example or other reasons…)?
7. Why did you only consider the longitudinal strain for 2D-SPE and not also the radial and circumferential strain (as you did for 3D-SPE), although these strains have already been described as interesting parameter of LV function? Have you tried to study the value of LV twist which is an even more integrative parameter of LV function?
8. How many operators have performed SPE measurements? As you evaluated the prognostic accuracy of 2D and 3D-SPE parameters, it is of importance to provide the intraobserver and interobserver reproducibility of SPE measurements as well as the least significant change for SPE measurements.
9. Please clearly indicate in Table 1 what are the 2D and the 3D-SPE variables.
10. Regarding the analysis of the prognostic value of the different echocardiographic parameters using ROC curves analysis, I guess that the different threshold values that are mentioned in Table 3 correspond to the best threshold values obtained after ROC curves analysis rather than to pre-specified cut-off values, as mentioned in the caption of the Table. Please clarify.
11. Please add a Figure with the most important ROC curves and with the corresponding individual data of the patients.
12. The logistic regression is confusing for readers, since “logistic regression” implicitly means multivariate analysis. It should be clearly stated in the “statistical analysis” section that you only performed univariate analysis. In addition, how did you select the pre-specified cut-off values for the different variables included in the model? Finally, you should have indicated in the current Table 4 that this was the unadjusted odds ratio.
13. Rather than performing a univariate logistic regression analysis, it would have been more appropriate and interesting to clearly assess the added-value of the SPE parameters over the more classical parameters by using for instance the net reclassification index.
14. I completely agree with the authors that the sample size is too small to perform a multivariate logistic regression analysis. Thus, because only a univariate analysis was performed, you should temper your message and conclusions regarding the association between LV function parameters and AF in patients with HCM. Please change accordingly the manuscript whenever necessary.
15. Please correct all the spelling errors throughout the manuscript.
16. Please define all the abbreviations used in the manuscript (for instance RVID).
Author Response
Response to the Second Reviewer
The authors wish to thank for the professional and most valuable comments
provided by the Reviewer. The Reviewer introduced critical new aspects and allowed for
the authors to approach their data with ample criticism. The authors took care to answer all
suggestions and remarks with due attention and diligence, to the limit of their study data.
Questions 1 and 2.
- The main goal of the study in the abstract (and in the manuscript) is not clear. What does “the discriminatory power of this arrhythmia” mean? Please clarify.
- The following sentence in the introduction is not clear: “Therefore, the role of LV assessment in this issue, not only the LA, cannot be overestimated”. Please clarify.
The authors thank the Reviewer very much for the critical questions and suggestions concerning the clarification mentioned above information. We completely agree with the Reviewer that the primary goal of the study is not clear. We tried to clarify that in the Introduction section and the Abstract.
So far, only LA diameter (greater than 45 mm) has been mentioned in cardiological guidelines as the only risk factor recommended in revealing AF in HCM patients, but the sensitivity and specificity of this parameter are too low; therefore, other indices of LA are intensively assessed in this issue. Recently, the evaluation of LV function, not only LA, in AF prediction enjoys growing interest. The connection between the LV and LA function is well-known, which seems to be particularly important in HCM patients. The pathophysiology of AF due to complex myocardial mechanics of LV and LA is undoubtedly different and more complex than in persons without this cardiomyopathy. Therefore, the role of LV assessment in AF prediction in HCM patients, additionally to LA assessment, seems to be of great importance. In this issue, 2D STE is a well-proven technique; however, almost none of the studies were dedicated to the role of LV strains in AF probability.
Moreover, strains calculated in three-dimensional (3D) STE could have a complementary role in HCM patients' assessment due to the potential ability to overcome some of the intrinsic limitations of 2D STE in assessing complex LV myocardial mechanics offering additional deformation parameters. Our study aimed to explore the value of LV assessment, using modern 2D and 3D echocardiography techniques in revealing the history of AF in HCM patients. We tried to answer whether the thorough LV assessment additionally to LA boosts the probability of AF diagnosis.
We tried to change the text in the manuscript (page 1-2, lines 79-82, 105-118, 124-128) and the abstract (page 1 Lines 53-56) clarifying our intention, for further Reviewer’s acceptance.
- Please describe what RVID is and how you have obtained this parameter.
The authors thank the Reviewer very much for that important question. Right ventricle internal diameter (RVID) was obtained and measured at end-diastole from the RV-focused 4-chamber view in the basal of RV inflow. We added necessary information in the Methods section (Page 3, line 191-192).
- Please clearly indicate the inclusion criteria of patients with HCM. When were the healthy volunteers included? Why did you include healthy volunteers? I do not quite understand what the added value of the healthy volunteers is in your analysis. I would suggest removing all data concerning the healthy volunteers. Please justify and clarify this point.
The authors are thankful to the Reviewer for this essential comment.
Firstly, according to the Reviewer's suggestion, the methods section has been modified to define inclusion criteria better, and it now reads as follows: “The HCM diagnosis was based on the current criteria: determined by echocardiography, at least one or more LV segmental hypertrophy ≥ 15mm, which could not be the result of abnormal loading condition (such as hypertension or aortic stenosis)” (Methods Section, page 2, lines: 133-135)
Secondly, regarding the remark about including the healthy volunteers group. Due to the lack of validated standards for 2D and especially 3D STE measurements, the authors intended to constitute healthy volunteers as a reference group for the STE measurements. Additionally, answering the other Reviewer’s question concerning the possible effect of the low LA volume on the AF, Table 2 presents that HCM patients had significantly higher LAVi.
Due to that facts, we added healthy volunteers to comparative analysis and added the necessary explanation into the text (Methods section, page 2, lines 142-143). The authors can follow the Reviewers insightful suggestion regarding removing all data concerning the healthy volunteers whenever necessary.
- How many patients were excluded because of poor echogenicity, preventing SPE analysis? Please also indicate in the manuscript and in Figure 1 the proportion of patients in whom 2D and 3D-SPE analysis were not possible.
Regarding the number of excluded patients with poor echogenicity, the authors are incredibly thankful for this valuable comment.
In our analysis, 88 HCM patients met the final clinical criteria; unfortunately, 28 of them had to be excluded due to the poor echogenicity enabling to perform speckle tracking analysis. The 3D - STE analysis requires good quality echocardiographic images more restrictive due to the necessity for an appropriate visualization of the hypertrophied myocardium for optimal tracking. The accurate measurement of the 2D LV GLS was able to perform in every included patient; the reliable analysis of LA strain was possible in 43 patients (due to the fact that we must excluded from our analysis LA segments with pulmonary veins and patients with interatrial septal aneurysm), whereas 3D LV strains were accurately measured in 45 patients. We added this valuable information in the Results section (page 6, lines: 273-275).
We kindly apologize to the Reviewer if our original Figure 1 did not clearly show the patients group excluded due to the insufficient echocardiographic images. The authors have added a specific explanation in the Methods section (Page 2, line: 141) in the Result section (page 6, lines: 273-275) and hope that now it is better expounded.
- Regarding SPE measurement, you have used the values automatically displayed by the software. How did you take into account the segments that were not adequately tracked (because of poor echogenicity for example or other reasons…)?
The authors wish to thank the Reviewer for this valuable question. The software permits the artifacts in at most two analyzed segments. Three or more segments with low tracking quality were automatically excluded from the analysis. Adequate tracking was verified in real-time and corrected by adjusting the region of interest or manually to ensure optimal tracking.
The authors underlined and developed this issue in Methods Section (page 3, lines: 199-203)
- Why did you only consider the longitudinal strain for 2D-SPE and not also the radial and circumferential strain (as you did for 3D-SPE), although these strains have already been described as interesting parameter of LV function? Have you tried to study the value of LV twist which is an even more integrative parameter of LV function?
The authors wish to thank the Reviewer for this valuable remark.
Firstly, the authors want to highlight that the mechanisms of rotation and twist, particularly in patients with HCM compared to healthy, are more complex and unique.
Additionally, we must admit that we performed the initial analysis of the rotation parameters. Also, since those parameters were unfortunately not statistically significant, we did not put them into the result section and did not perform further analyses. Thank the Reviewer’s comment; we have already placed the twist and torsion measurements in Table 2 (page 7) and proper clarification in Discussion (page 11, lines: 413-418) section.
Secondly, according to the remark about not considering 2D circumferential and radial strain, we should admit that we were more concentrated on simple and easy to perform parameters; therefore, we included only 3D STE measurements, as they were possible to obtain from one analysis. For the time being, a sentence clearly stating this drawback was added to the “study limitations” section (page 12 lines 500-502). We hope that the Reviewer will accept this answer.
- How many operators have performed SPE measurements? As you evaluated the prognostic accuracy of 2D and 3D-SPE parameters, it is of importance to provide the intraobserver and interobserver reproducibility of SPE measurements as well as the least significant change for SPE measurements.
Regarding the remark about the reliability and reproducibility of performed STE measurements, we are thankful for this valuable question. The authors entirely agree that it is essential to provide the inter and intraobserver reproducibility of STE values. We added the appropriate analyses concerning the reliability and reproducibility in the Statistics section (page 4, lines: 242-246), and included the results (Results section, page 10, lines: 355-360, Table 5); also, the precise results presents Table 5 Two experienced echocardiographers performed all analyses of echocardiographic parameters, and we added this information in the Methods section (page 3, line 177)
- Please clearly indicate in Table 1 what are the 2D and the 3D-SPE variables.
The authors wish to thank for this valuable remark. We fully agree with the Reviewer and have already made the appropriate changes in Table 2 (page 7).
- Regarding the analysis of the prognostic value of the different echocardiographic parameters using ROC curves analysis, I guess that the different threshold values that are mentioned in Table 3 correspond to the best threshold values obtained after ROC curves analysis rather than to pre-specified cut-off values, as mentioned in the caption of the Table. Please clarify.
Regarding the valuable Reviewers remark about the analysis of the predictive value of the different echocardiographic parameters using ROC curves analysis, we want to thank the Reviewer for this reliable comment. The authors fully agree with the Reviewer and wish to apologize for did not state this clearly. We add the appropriate changes in line with the Reviewers suggestion (page 8-9 Table 3 and page 9 line 345-346).
- Please add a Figure with the most important ROC curves and with the corresponding individual data of the patients.
We want to thank the Reviewer for this reliable comment. Due to the valuable Reviewers suggestion, the authors have already inserted the appropriate Figure, which presents essential ROC curves (page 9, Figure 6). We hope that the Reviewer will accept this Figure.
- The logistic regression is confusing for readers, since “logistic regression” implicitly means multivariate analysis. It should be clearly stated in the “statistical analysis” section that you only performed univariate analysis. In addition, how did you select the pre-specified cut-off values for the different variables included in the model? Finally, you should have indicated in the current Table 4 that this was the unadjusted odds ratio.
Regarding the remark about the meaning of “logistic regression”, the authors wish to thank for this comment. We completely agree with the Reviewer. We changed the phrase to the correct one as suggested (Statistical Section, page 4, line 161). Additionally, the authors intend to explain selecting the pre-specified cut-off values for the different variables included in the model. According to the previous remark, those values were determined using the area under the receiver-operating characteristic (ROC) curve (AUC). Secondly, we performed univariate logistic regression analysis to search the strongest relation of analyzed parameters to the presence of AF. We corrected the Table 4 abbreviation to indicate the statistical method appropriately (page 9, line 348). The authors hope this answer will be acceptable for the Reviewer.
- Rather than performing a univariate logistic regression analysis, it would have been more appropriate and interesting to clearly assess the added-value of the SPE parameters over the more classical parameters by using for instance the net reclassification index.
The authors agree that the use of the net reclassification index as an interesting statistical analysis would be of value. However, mainly due to the small sample group, we were not able to perform more advanced and complex analysis concerning other classical parameters (apart from LADs, which had lower than STE parameters AUC value, sensitivity, specificity, positive and negative predictive value), concentrating on echocardiographic parameters which significantly differed between AF+ and AF- groups. We would be very keen on carrying out such analysis, possibly on a more significant number of patients, in the future. For the time being, a sentence clearly stating this drawback was added to the “study limitations” section (page 12, lines 498-500). The authors hope that the Reviewer will accept this answer.
- I completely agree with the authors that the sample size is too small to perform a multivariate logistic regression analysis. Thus, because only a univariate analysis was performed, you should temper your message and conclusions regarding the association between LV function parameters and AF in patients with HCM. Please change accordingly the manuscript whenever necessary.
The authors wish to thank for this valuable comment. We fully agree with the Reviewer's remark that because we performed only univariate analysis, we should temper our message and conclusion regarding the association between LV function parameters and AF occurrence in patients with HCM. The authors changed the matter accordingly (page 10: lines: 365-366 page 11: lines: 407, 410, 412, 416, 427-428, 435) for the further Reviewers acceptance.
- Please correct all the spelling errors throughout the manuscript.
According to the Reviewers remark, the authors tied to correct the spelling errors throughout the manuscript.
- Please define all the abbreviations used in the manuscript (for instance RVID).
The authors wish to thank the Reviewer for pointing this out. The authors revised and corrected all the abbreviations used in the manuscript for

Round 2
Reviewer 1 Report
REVISED Manuscript ID diagnostics-1287013 entitled "Comprehensive echocardiography of left atrium and left ventricle using modern techniques helps better revealing atrial fibrillation in patients with hypertrophic cardiomyopathy."
Authors answered to general comments of this reviewer. I have no further comments.
Reviewer 2 Report
The authors have taken into account all the comments of the reviewers and have significantly improved their manuscript. I have no additionnal comment.